# When election expectations fail: Polarized perceptions of election legitimacy increase with accumulating evidence of election outcomes and with polarized media

**Marrissa D. Grant[1], Alexandra Flores[1], Eric J. Pedersen[1], David K. Sherman[2], Leaf Van Boven**[1]*

**1** University of Colorado Boulder, Boulder, Colorado, United States of America, **2** University of California, Santa Barbara, Santa Barbara, California, United States of America

* vanboven@colorado.edu

**Data Availability Statement:** All data, materials, and code available at: DOI 10.17605/OSF.IO/EWR7G.

## Abstract

The present study, conducted immediately after the 2020 presidential election in the United States, examined whether Democrats' and Republicans' polarized assessments of election legitimacy increased over time. In a naturalistic survey experiment, people ($N$ = 1,236) were randomly surveyed either during the week following Election Day, with votes cast but the outcome unknown, or during the following week, after President Joseph Biden was widely declared the winner. The design unconfounded the election outcome announcement from the vote itself, allowing more precise testing of predictions derived from cognitive dissonance theory. As predicted, perceived election legitimacy increased among Democrats, from the first to the second week following Election Day, as their expected Biden win was confirmed, whereas perceived election legitimacy decreased among Republicans as their expected President Trump win was disconfirmed. From the first to the second week following Election Day, Republicans reported stronger negative emotions and weaker positive emotions while Democrats reported stronger positive emotions and weaker negative emotions. The polarized perceptions of election legitimacy were correlated with the tendencies to trust and consume polarized media. Consumption of Fox News was associated with lowered perceptions of election legitimacy over time whereas consumption of other outlets was associated with higher perceptions of election legitimacy over time. Discussion centers on the role of the media in the experience of cognitive dissonance and the implications of polarized perceptions of election legitimacy for psychology, political science, and the future of democratic society.

## Introduction

Healthy democracies rest on shared confidence in election legitimacy [1]. Citizens need to regard elections as fair and legitimate for governmental effectiveness. Particularly important, yet also particularly challenging, is that people whose preferred candidate lost an election

**Funding:** This work was supposed by a grant to LVB from the National Science Foundation SES: 2029183. The funders had no role in study design, data collection and analysis, decision to publish, or preparation of the manuscript.

**Competing interests:** The authors have declared that no competing interests exist.

nevertheless accept the outcome as legitimate [2–4]. "Loser's consent" is an indicator of a well-functioning democracy. Without such consent, widespread questioning of election legitimacy may reduce trust in government, catalyze mass protest, and trigger violence [5]. Americans witnessed all three outcomes following the 2020 election of President Joseph Biden and the defeat of former President Donald Trump. What does social psychology suggest about the public's perception of election legitimacy?

We take our theoretical inspiration for this study, a naturalistic experiment during the 2020 presidential election in the United States, from classic research in social psychology and the research tradition spurred by cognitive dissonance theory [6, 7]. A recent evaluation of Leon Festinger and cognitive dissonance theory's contributions to social psychology and society written by Lee Ross [8] makes several historical and theoretical points that anticipated our study and highlights its relevance. Ross notes a dearth of research inspired by cognitive dissonance theory that examines people's reactions to real-world events where strong feelings of dissonance are widely experienced. A limitation of the traditional dissonance methodology, Ross notes, was that "the levels of dissonance experienced by the participants in most studies (generally college undergraduates or children) were much lower than the levels experienced by individuals who had faced soul-challenging decisions. . . (p. 9)" A tightly contested election, one where each side had strongly held, identity-intertwined cognitions and viewed it as a "soul-challenging decision" for the nation would seem to provide a naturalistic laboratory to examine dissonance phenomena. Ross writes:

> . . . real-world field research received little attention. One such research target would have been assessments of the merits of political candidates immediately after versus before voting, *or before versus after election results become known*. . . However such phenomena seem to have been left to political scientists. [8, p. 10, emphasis added]

Indeed, political scientists have examined the phenomenon whereby partisans from the losing party perceive elections as less legitimate than partisans from the winning party [9–12]. Although political scientists have linked polarized perceptions of election legitimacy to theories such as cognitive dissonance theory [6, 13–17], the methods tend to confound before versus after voting with before versus after results become known because those two factors are closely intertwined in most elections.

The experience of a national election consists of a torrent of information and outcomes for people considering a consequential choice: the candidate for whom they voted. Before an election, both sides hope for and often expect their candidate to win. As election outcomes become known, dissonance can be evoked by conflicting cognitions people feel in the wake of a national election, such as the discrepancy between one's view that that "my country is a decent place" and "my country elected a horrible person."

In this paper, we focus on the discrepant feelings people experience during and after the election outcome as it relates to their preferred candidate. For those on the losing side, the negative outcome and unexpected disappointment stemming from their candidate's loss is dissonant with their perceptions of the candidate's positive attributes, which were insufficient to compel a winning majority. One way this can be accomplished is by questioning the legitimacy of votes, suggesting that their candidate's loss is illegitimate and that, if votes had been counted correctly, the candidate would not have lost. For winners, the expected positive outcome of their candidate's win is dissonant with any lingering doubts about their candidate's ability to govern effectively and other perceptions of the candidate's weakness. Reducing this dissonance can be done by affirming the legitimacy of votes, thereby bolstering the candidate's widespread appeal and legitimacy of the candidate's win. Theories of dissonance-induced rationalization

thus imply that polarized perceptions of election legitimacy should increase over time as the election outcomes become widely known [18].

We report the results of a naturalistic survey experiment conducted during the 2020 presidential election in the United States. We tested whether Democrats, whose presidential candidate Joseph Biden won the election, would perceive the election outcome as more legitimate, as indicated by confidence that votes were correctly counted, than Republicans, whose candidate Donald Trump lost the election. We also tested the crucial prediction that these polarized differences would increase over the two weeks following Election Day, as votes were counted, outcome certainty increased, and President Biden was widely declared the winner.

The present study takes advantage of the unique circumstance of the 2020 presidential election in the United States to isolate knowledge of election outcome from the possibility of voting or influencing votes. The present study also provides indirect correlational evidence for the role of social confirmation through the consumption of polarized media in reducing dissonance by increasing polarized perceptions of election legitimacy. We do so by examining whether trust and consumption of media outlets that voiced skepticism about election legitimacy (i.e., Fox News) was correlated with decreased perceptions of election legitimacy as vote counts and the election outcome became more widely known.

## Polarized perceptions of election legitimacy

Previous research in political science demonstrates that the tendency for winners to perceive elections as more legitimate than losers increases from before to after elections, consistent with predictions derived from cognitive dissonance theory [18]. However, those studies necessarily confound whether people have learned the election outcome with the possibility of voting in the election [9–12], leading to theoretical imprecision in what drives polarized perceptions of election legitimacy. Before an election, citizens can still influence election outcomes by voting and convincing others to vote for their candidate. After an election, citizens have learned the outcome, typically announced the evening of Election Day in the United States, and they can no longer cast or influence votes. Of course, as the unfolding of the 2020 American presidential election revealed, people could also question the legitimacy of the voting process after its conclusion through a multitude of actions such as legal disputes about election practices or disrupting the certification of voting outcomes. Nevertheless, there is a qualitative difference between these highly unusual actions and the sanctioned actions people can take to influence voting before an election day. Pre- to post-election comparisons that have been typically featured in research confound two plausible causes of polarized perceptions: outcome knowledge and opportunities to influence elections.

Disentangling this confound is important because the mere act of voting may increase polarized attitudes [19–21]. In one study, voters entering a polling station, who had not yet voted, held less favorable opinions of their candidate than voters who were exiting the polling station, having just cast their vote [19]. Without knowing the election outcome, the act of voting was sufficient to increase polarized perceptions of the candidate. This raises the possibility that increasingly polarized perceptions of election legitimacy from pre- to post-election is due to people having voted, not to their having learned the election outcome.

The 2020 U.S. presidential election presented an opportunity to disentangle these interpretations. The election had a historically high turnout, an unprecedented number of mail-in and absentee ballots, and social distancing practices during the COVID-19 pandemic. This contributed to a period following Election Day when votes were cast but the winner was uncertain. Four days after Election Day, on November 7, 2020, Biden was widely declared the winner (Fig 1). Comparing perceptions of election legitimacy during the contested, undeclared period

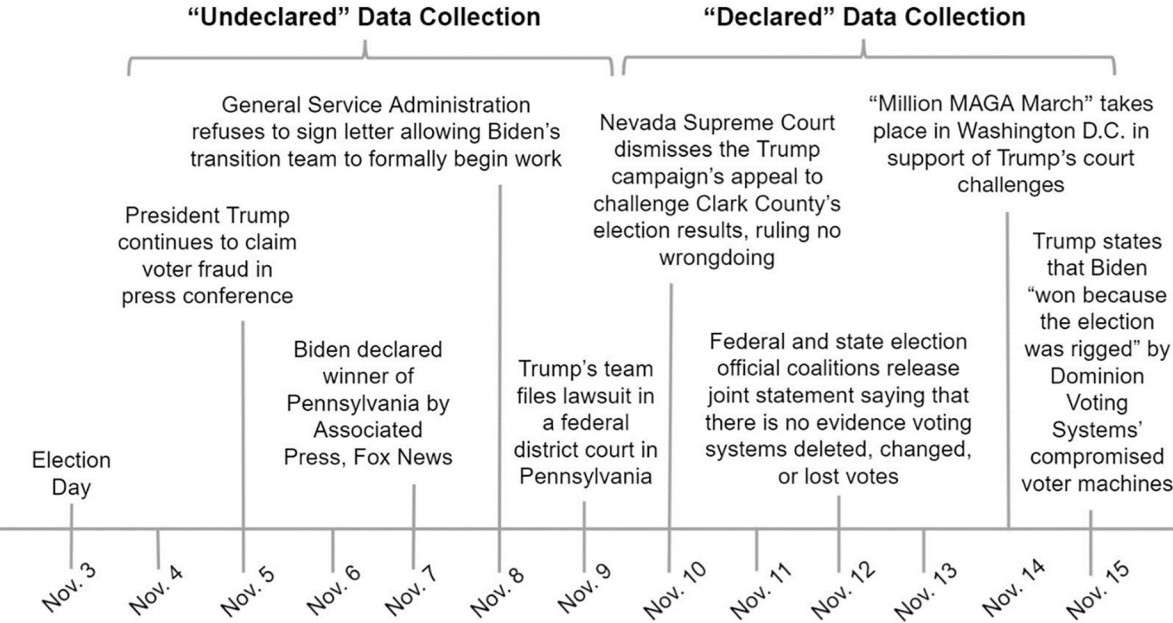

**Fig 1. Timeline of election events over course of data collection.**

with perceptions after the outcome was widely declared removes the possible impact of individuals influencing the actual vote total (by either voting or encouraging others to vote), thus affording greater theoretical precision in testing the influence of learning the election outcome. With widespread predictions that the 2020 U.S. presidential election would not be called on election night, we designed a study to disambiguate these interpretations, and moreover, to examine correlated emotions and the correlated consumption of and trust in polarized media.

### Emotional election outcomes

We also examined whether the psychological dynamics of a presidential election would provoke associated positive and negative emotions. Because partisans are personally invested in election outcomes, their group-based emotional experience should reflect emerging knowledge that their favored candidate won or lost the election [22, 23]. Negative feelings of anger, anxiety, irritability, and nervousness are indicators of disappointment and, we reason, the discomfort stemming from any cognitive dissonance elicited from an unexpected election loss [24]. Positive emotions of hope, happiness, and excitement are indicators of the pleasure of a hoped-for but uncertain election win. Given that President Biden won the election, we examined whether Democrats would exhibit subsiding negative emotions while Republicans would exhibit intensifying negative emotions, with positive emotions following the reverse pattern.

### Polarized media, social confirmation, and election legitimacy

Elections are social events, so it is important to also examine dissonance reduction processes within the broader social context that occurs during and after elections. The arousal and reduction of cognitive dissonance is often a social process [7]. Yet much research inspired by cognitive dissonance theory has focused on the internal dynamics and attitudinal consequences of having engaged in self-relevant, seemingly freely chosen, counter-attitudinal behavior [25]. In his essay on cognitive dissonance theory, Ross noted that "perhaps the most

important shortcoming of the dissonance theory tradition was its almost exclusive focus on individual rather than collective processes" (p. 15).

The experiences of Democrats and Republicans in the wake of the 2020 presidential election as they made sense of (and rationalized) the outcome were likely influenced by the collective processes reflected by partisan media. Major media outlets and their associated information ecosystems both convey information about the reactions of fellow Democrats and Republicans, and provide rationalizing information consistent with dissonance reducing claims about the election [26, 27]. Partisan media may serve the social function of reflecting and abetting polarized beliefs [28–31].

Reducing cognitive dissonance is one means towards restoring and maintaining cognitive consistency [32–34]. The influence of partisan media in polarizing perceptions of election legitimacy should therefore depend on engagement with consistent, homogeneous media that provide coherent, confirmatory information. Consistent with this analysis, liberals and conservatives with more extreme stances exhibit greater homogeneity in their social media networks [35], which may similarly occur with mainstream media outlets.

In the context of U.S. politics, Fox News is singularly trusted by Republicans and distrusted by Democrats and Independents [36, 37]. Unlike other major mainstream media outlets, Fox News stood out in questioning the legitimacy of the 2020 U.S. presidential election, asking whether votes were correctly counted or fraudulent [38, 39]. In this way, partisan media may provide evidence to support individuals' polarized perceptions of election legitimacy, even in the face of accumulating evidence confirming vote counts and the election outcome. We thus examined whether increased polarized perceptions of election legitimacy followed engagement with partisan media. We specifically examined whether trust in and consumption of mainstream media outlets was associated with increased perceptions of election legitimacy whereas trust and consumption of Fox News was associated with reduced perceptions of election legitimacy.

## Study overview

We examined polarized perceptions of election legitimacy in a naturalistic experiment following the 2020 U.S. presidential election. Because it was widely anticipated that the election would not be called on election night, we randomly assigned participants from a larger sample that had participated in earlier research related to perceptions of COVID-19 policies and the media, to one of two conditions: People either completed a survey study during the week following Election Day, with votes cast but the outcome still to be determined, or the following week, when Biden was the widely declared winner (Fig 1). We defined the week following Election Day (November 4–8) as "Undeclared" because many states' vote counts had not been confirmed and the outcome was unknown by everyone, particularly during the days immediately following election day. We defined the following week (November 9–15) as "Declared" because all states' vote counts had been confirmed, the Associated Press (and most other media outlets, including Fox News, if not its prominent commentators) had declared President Joe Biden as winner, and he had delivered his acceptance speech. The time frame covered by our study obviously reflected a continuous process of new information and events. Nevertheless, the dividing line reflects a period when no one knew who was declared winner versus a period when most people presumably knew that Biden was declared winner.

Thus, the design was not a cross-sectional design whereby participants were selected at two time points, but a true experiment, where participants had an equal chance of being either in the "Undeclared" or the "Declared" period. The study, however, has elements of a natural experiment, in that the independent variable ("Undeclared" vs. "Declared" election outcome)

depended on events occurring in the world outside of the researchers' control: When the victor of the 2020 U.S. presidential election was publicly declared by the major media outlets.

We used an established measured of perceived election legitimacy, operationalized as confidence that votes were counted as intended [3, 10]. We also measured participants' expectations about who would win the election. We examined whether Democrats and Republicans would expect their candidate to win, whether Democrats would perceive the election as more legitimate than Republicans, and whether these polarized perceptions of election legitimacy would increase over time.

We also measured people's emotional reactions to the election. From the Undeclared to Declared periods, we examined whether Democrats' negative emotions would subside while positive emotions would increase, and whether Republicans' negative emotions would increase while positive emotions would subside. We further examined the associations between these emotional reactions and perceptions of election legitimacy.

Finally, we asked participants to report how much they trusted and consumed Fox News and 14 other widely consumed media outlets including CNN, The New York Times, and The Wall Street Journal. Confirming other research, we expected Republicans to trust and consume Fox News more than Democrats, who would trust and consume other media outlets [37, 40]. We explored whether trust and consumption of Fox News and of other media would independently predict and moderate perceptions of election legitimacy and changes over the two-week period.

## Method

The Institutional Review Board at the University of Colorado categorized the study as Exempt (Protocol 20–0197). All data, materials, and code are available at https://osf.io/ewr7g/.

### Participants

Participants were U.S. residents (*N* = 1,236, 44.5% female, 47.2% male, with the remaining other/unspecified), recruited using ROI Rocket, and paid $4. The sample was diverse in age ($M_{age}$ = 49.71; $SD_{age}$ = 15.21, range [18, 89]) and ethnicity (11.7% African American, 8.2% Asian American, 6.8% Latin American, 62.7% White, and 11.8% other or declined to provide). As in previous research [e.g., 41], we measured partisan identification using a two-step procedure with dichotomous questions from the American National Election Study. This allowed categorization, including leaners, of participants as Democrat (*N* = 565), Republican (*N* = 466), or Independent (*N* = 194), with 11 who did not report their partisan identification.

The sample yielded an 80% chance to detect an effect of F = 3.0 and the sample size was determined based on available funding. Sample sizes differ across analyses due to missing data.

### Procedure

A total of 1,672 potential participants were randomly assigned to one of two survey periods, either November 4–8 or November 9–15, 2020. Response rates for both periods (583 of 836 = 70% during the Undeclared period; 653 of 836 = 78% during the Declared period) were consistent with online panel response rates [42] and were not significantly different from each other (*p* = .134). We referred to the first period as the "Undeclared" period because all votes had been cast but not fully counted leaving the election outcome unknown and not declared.

Anticipating a delay between election night and declared election results, we planned to leave the Undeclared group's survey open until a winner was officially called; we closed the first group's survey on November 8th, one day after the Associated Press called the election for Biden. We then launched the survey for the second, Declared group on November 9th. We referred to the second period as "Declared" because Joe Biden had delivered his acceptance

speech and had been widely declared to have won the election. Of our Undeclared sample (*N* = 583), 17 people participated during the Declared period (because they participated on November 8[th]). Excluding these people does not substantively impact the pattern of results (see OSF); of course, including them as we did works against the hypotheses.

## Measures

**Expected outcome.**   We measured participants' expected election outcome with a single item, with wording in brackets for participants in the declared period: "Who do [did] you think will [would] ultimately win the presidential election?" Responses were on an ordinal scale presented without numbers, with counterbalanced scale anchors (1 = *Definitely Donald Trump;* 5 = *Unsure/Toss-up;* 9 = *Definitely Joe Biden*).

**Perceived election legitimacy.**   We measured participants' perceived election legitimacy by averaging answers to two questions about confidence that votes were counted correctly. One item measured perceptions of own vote confidence and the other perceptions of nationwide vote confidence, respectively: "How confident are you that your vote in the general election was counted as you intended?" and "How confident are you that votes nationwide in the general election were counted as voters intended?" [3, 10]. Participants answered on two ordinal scales presented without numbers (1 = *Not at all confident*; 5 = *Very confident*). Responses were highly correlated (*r* = .75).

**Emotion.**   Participants reported their emotions about the presidential election: "When you think about the election, how much do you feel each of the following?" (1 = *Not at all*; 7 = *Extremely*). Negative emotions were averaged into an index comprising anger, guilt, shame, embarrassment, nervousness, distress, and irritability (α = .88). Positive emotions were averaged into an index comprising pride, gratitude, hope, happiness, and excitement (α = .93).

**Media trust and consumption.**   We measured participants' trust and consumption of media by asking people how much they trusted and consumed 15 media outlets: Fox News, ABC News, AOL News, CBS News, CNN, Huffington Post, MSNBC, NBC News, NPR, New York Times, PBS, USA Today, Washington Post, Wall Street Journal, and Yahoo News. We did not include a broader range of conservative media outlets such as One American News Network, Newsmax, or Breitbart because they lacked available and established full-text databases that we required for a project unrelated to the present manuscript regarding linguistic analysis of media content. Participants answered, "How much do you distrust or trust the accuracy of reporting for that source?" (1 = *Distrust completely*; 5 = *Trust completely*). We also asked them, as part of the larger research project on media consumption and COVID-19, "In general, how much do you get news about COVID-19 from each source?" (1 = *Not at all*; 5 = *A great deal*). Participants were presented with non-numeric scales when answering these items. For each participant, we computed the average correlation between trust and consumption ratings for each of the 15 media outlets. Within participants, the average *r* was .51 (*SD* = 0.33). Examined differently, for each media outlet, we calculated the correlation across participants of their trust and consumption ratings (the average *r* across 15 outlets was also equal to .51, range [.36, .67]). Patterns of significance are the same when we separately examine the measures of trust and consumption. Because previous research demonstrated that Fox News is uniquely trusted by Republicans compared with other media outlets [37, 40], we averaged participants' ratings of the 14 media outlets other than Fox News.

## Results

### Expected election outcome

Democrats and Republicans had different expectations of the election outcome, as reflected by a main effect of partisanship in a 3(partisan identification: Democrat, Republican, or

Independent) × 2(timing: Undeclared, Declared) ANOVA ($F(2, 1078) = 284.76$, $p < .001$, $\eta_p^2 = .359$). Compared with the scale midpoint of 5, Republicans expected Trump to win ($M = 3.63$, $SD = 2.64$; $F(1, 1078) = 143.35$; $p < .001$, $\eta_p^2 = .117$), Democrats expected Biden to win ($M = 7.42$, $SD = 2.02$; $F(1, 1078) = 484.57$, $p < .001$, $\eta_p^2 = .310$), with Independents in between ($M = 5.38$, $SD = 2.30$; $F(1, 1078) = 2.82$, $p = .093$, $\eta_p^2 = .004$). The interaction between partisan identification and timing was not significant ($F(2, 1078) = 2.20$, $p = .112$, $\eta_p^2 = .004$). Each side believed their candidate would win, meaning that after Biden was declared winner, Democrats' expectations were confirmed, and Republicans' expectations were disconfirmed.

## Perceptions of election legitimacy

Our central prediction was that polarized perceptions of election legitimacy would increase after Biden was the widely declared winner compared with the undeclared period when votes were cast but the outcome was unknown (Fig 2). To test this, we conducted a 3(partisan identification: Democrat, Republican, Independent) × 2(timing: Undeclared, Declared) ANOVA on the aggregate measure of perceived election legitimacy. The key predicted interaction between partisan identification and timing was significant ($F(2, 1202) = 10.80$, $p < .001$, $\eta_p^2 = .018$). From the Undeclared to Declared periods, Democrats became more confident that votes were counted as intended ($M_{\text{Undeclared}} = 3.92$, $SD_{\text{Undeclared}} = 1.00$; $M_{\text{Declared}} = 4.37$, $SD_{\text{Declared}} = 0.92$; $F(1, 1202) = 21.18$, $p < .001$, $\eta_p^2 = .017$), while Republicans became less confident that votes were counted as intended ($M_{\text{Undeclared}} = 2.99$, $SD_{\text{Undeclared}} = 1.18$; $M_{\text{Declared}} = 2.79$, $SD_{\text{Declared}} = 1.24$; $F(1, 1202) = 3.94$, $p = .047$, $\eta_p^2 = .003$), and Independents' confidence remained unchanged over time ($M_{\text{Undeclared}} = 2.96$, $SD_{\text{Undeclared}} = 1.28$; $M_{\text{Declared}} = 3.02$, $SD_{\text{Declared}} = 1.45$; $F(1, 1202) < 0.01$, $p = .983$, $\eta_p^2 < .001$).

The increase in Democrats' perception of election legitimacy was larger in magnitude than the decrease in Republicans' perception of election legitimacy. This may simply reflect that Democrats' increasing confidence that votes were counted correctly coincided with accumulating evidence from state certification of vote counts. Republicans' decreasing confidence that votes were counted as intended occurred despite official certification of vote counts.

## Emotions

The polarized perceptions of election legitimacy and expectations that were dashed or confirmed were accompanied by polarized emotions (Fig 3). A 3(partisan identification: Democrat, Republican, Independent) × 2(timing: Undeclared, Declared) × 2(emotion valence: negative, positive) ANOVA with repeated measures on the last factor revealed a 3-way interaction ($F(1, 1208) = 87.05$, $p < .001$, $\eta_p^2 = .067$).

For negative emotions, there was a significant interaction in a 3(partisan identification: Democrat, Republican, Independent) × 2(timing: Undeclared, Declared) ANOVA ($F(2, 1208) = 12.83$, $p < .001$, $\eta_p^2 = .013$). From the Undeclared to Declared periods, Republicans' negative emotions increased ($M_{\text{Undeclared}} = 2.93$, $SD_{\text{Undeclared}} = 1.39$, $M_{\text{Declared}} = 3.19$, $SD_{\text{Declared}} = 1.54$; $F(1, 1208) = 3.92$, $p = .048$, $\eta_p^2 = .003$) while Democrats' negative emotions decreased ($M_{\text{Undeclared}} = 3.35$, $SD_{\text{Undeclared}} = 1.62$, $M_{\text{Declared}} = 2.71$, $SD_{\text{Declared}} = 1.46$; $F(1, 1208) = 27.88$, $p < .001$, $\eta_p^2 = .023$) and Independents' negative emotions were unchanged ($M_{\text{Undeclared}} = 2.64$, $SD_{\text{Undeclared}} = 1.54$, $M_{\text{Declared}} = 2.54$, $SD_{\text{Declared}} = 1.47$; $F(1, 1208) = 0.18$, $p = .667$, $\eta_p^2 < .001$). Positive emotions revealed the inverse shift, as reflected by an analogous interaction ($F(2, 1208) = 40.43$, $p < .001$, $\eta_p^2 = .132$). From the Undeclared to Declared periods, Democrats' positive emotions increased ($M_{\text{Undeclared}} = 3.26$, $SD_{\text{Undeclared}} = 1.60$, $M_{\text{Declared}} = 4.66$, $SD_{\text{Declared}} = 1.75$;

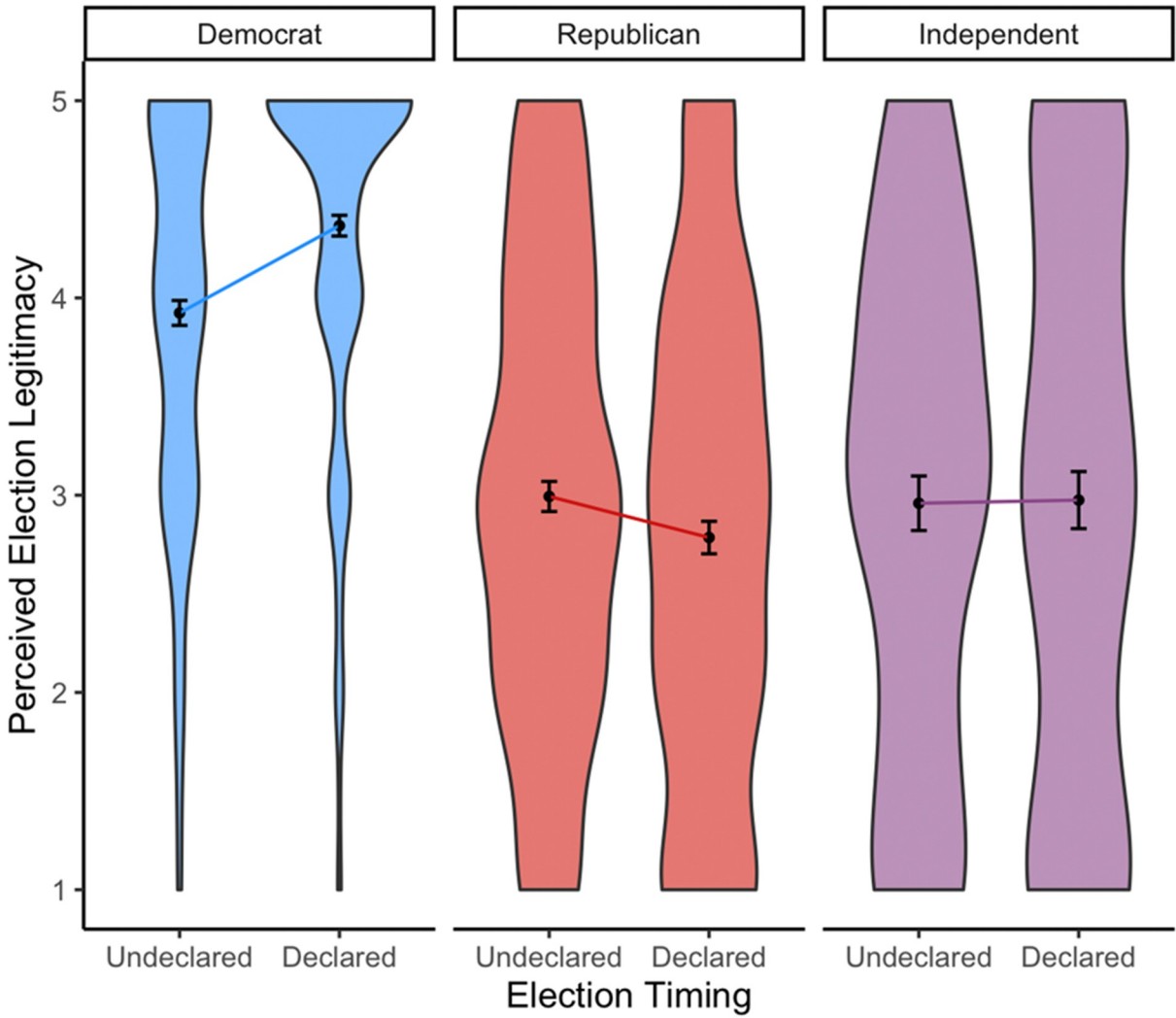

**Fig 2. Violin density plots, means, and +/−SE of perceived election legitimacy among Democrats, Republicans, and Independents who reported confidence that both their own and nationwide votes were counted as intended.** The width of the density plot represents the relative portion of each sample at each value of election legitimacy perception.

$F(1, 1208) = 83.54$, $p < .001$, $\eta_p^2 = .065$) while Republicans' positive emotions decreased ($M_{\text{Undeclared}} = 3.11$, $SD_{\text{Undeclared}} = 1.65$, $M_{\text{Declared}} = 2.61$, $SD_{\text{Declared}} = 1.73$; $F(1, 1208) = 11.83$, $p = .0006$, $\eta_p^2 = .010$) and Independents remain unchanged ($M_{\text{Undeclared}} = 2.44$, $SD_{\text{Undeclared}} = 1.47$, $M_{\text{Declared}} = 2.25$, $SD_{\text{Declared}} = 1.60$; $F(1, 1208) = 0.24$, $p = .628$, $\eta_p^2 < .001$).

These emotional reactions were associated with perceptions of election legitimacy. In a multiple regression analysis, negative emotions were associated with lower perceptions of election legitimacy ($b = -0.12$, $F(1, 1200) = 30.03$, $p < .001$, $\eta_p^2 = .024$) whereas positive emotions were associated higher perceived election legitimacy ($b = 0.19$, $F(1, 1200) = 105.70$, $p < .001$, $\eta_p^2 = .081$), controlling for partisan identification ($F(2, 1200) = 122.20$, $p < .001$, $\eta_p^2 = .169$), timing ($b = 0.02$, $F(1, 1200) = 0.07$, $p = .799$, $\eta_p^2 < .001$), and their interaction ($F(2, 1200) = 0.94$, $p = .390$, $\eta_p^2 = .002$). To the extent that people perceived the election as legitimate, they reported more positive emotion and less negative emotion.

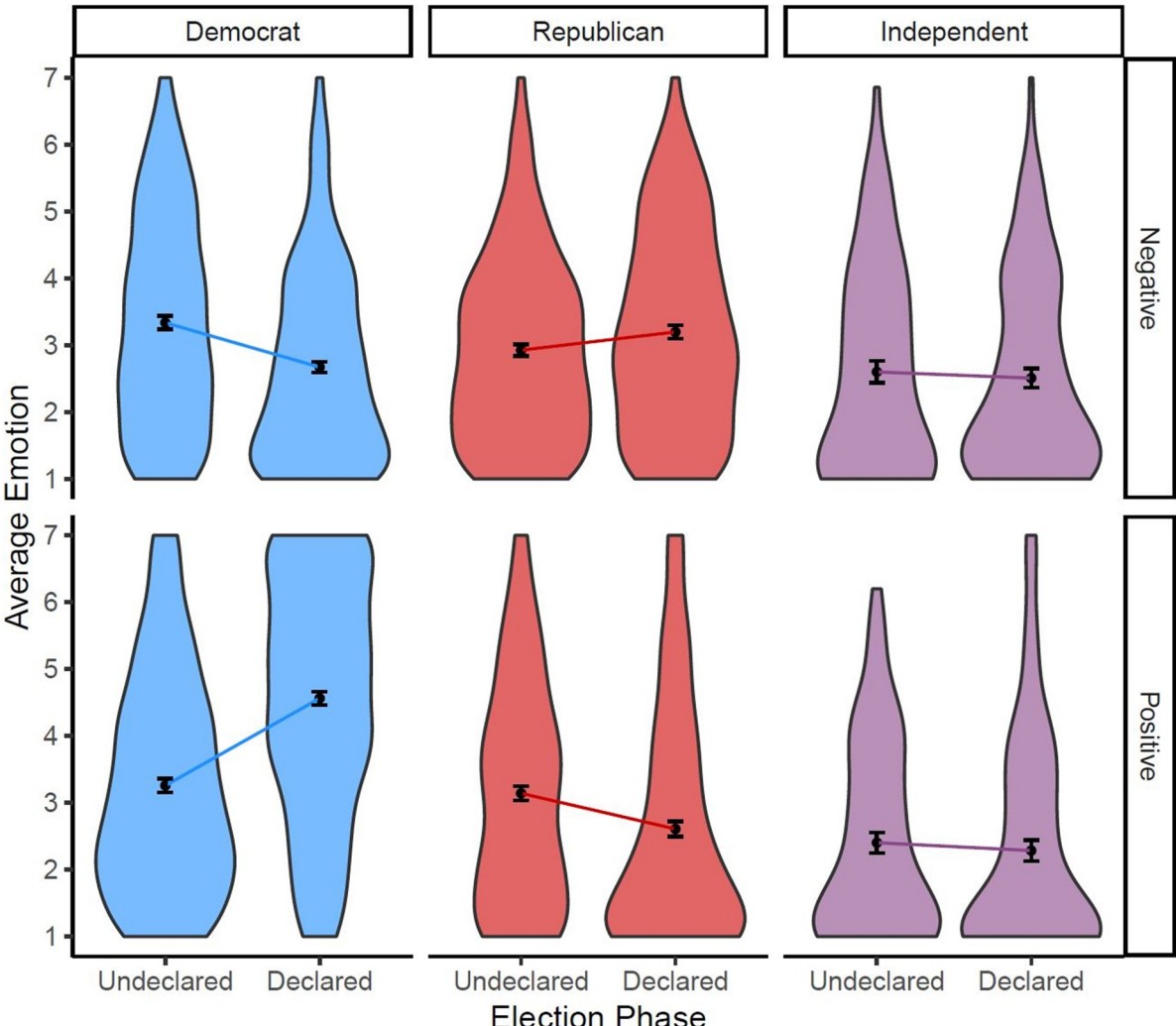

**Fig 3. Violin density plots, means, and +/−SE of average emotion, negative in the top panel and positive in the bottom panel, among Democrat, Republican, and Independent participants during two weeks during and after the 2020 presidential election.** Participants reported their negative emotions (anger, shame, embarrassment, nervousness, distress, and irritability) and positive emotions (pride, gratitude, hope, happiness, and excitement). The width of the density plot represents the relative portion of each sample at each value of negative and positive emotion.

## Media trust and consumption

Democrats and Republicans were polarized in their trust and consumption of media outlets (Fig 4). A 3(partisan identification: Democrat, Republican, Independent) × 2 (election timing: Undeclared vs. Declared) ×2(media outlet: Fox News, Other Outlet) ANOVA with repeated measures on media outlet revealed a nonsignificant interaction between partisan identification and media outlet ($F(2, 1206) = 37.00$, $p < .001$, $\eta_p^2 = .329$). However, Republicans reported trusting and consuming Fox News ($M = 2.96$, $SD = 1.28$) more than did Democrats ($M = 1.97$, $SD = 1.10$, $F(1, 1209) = 185.91$, $p < .001$, $\eta_p^2 = .133$), who reported trusting and consuming the 14 other outlets ($M = 2.89$, $SD = 0.66$) more than did Republicans ($M = 1.99$, $SD = 0.84$; $F(1, 1209) = 357.59$, $p < .001$, $\eta_p^2 = .228$). Independents' ratings of media outlets were in between Democrats and Republicans. The 3-way interaction was not significant ($F(2, 1206) = 1.03$, $p =$

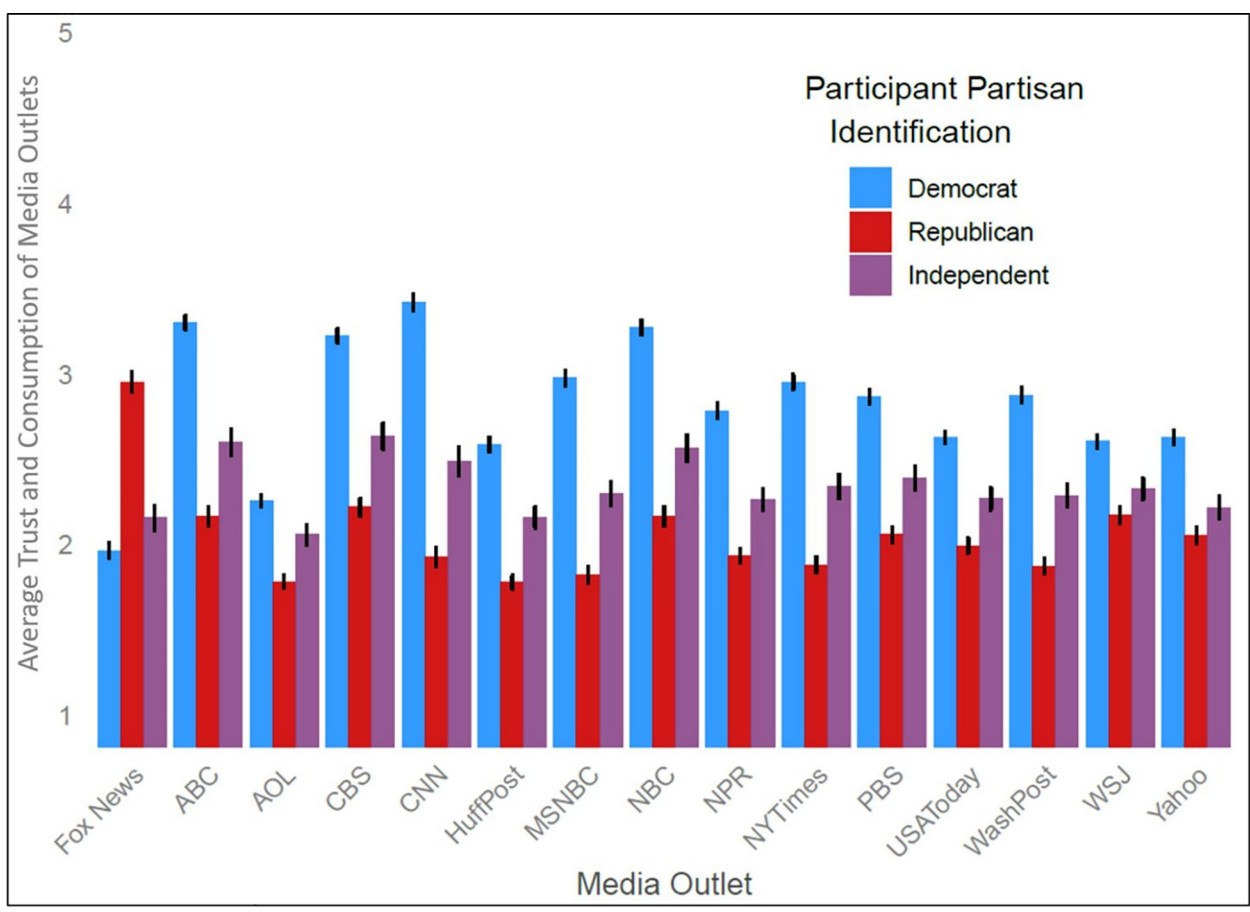

**Fig 4. Mean and–/+ SE of media trust and consumption of 15 major media outlets, the average of participants' trust in each outlet and their reported consumption of COVID-19 news from each outlet.**

.356, $\eta_p^2$ = .002), indicating that this pattern of results did not change significantly between the undeclared and declared periods. These results confirm that Democrats and Republicans trust and consume different media outlets, with Fox News being uniquely highly rated by Republicans.

To explore these relationships more extensively, we regressed the measure of perceived election legitimacy on participant partisan identification, election timing, ratings of Fox News, ratings of other media outlets, and their interactions (Table 1). Consistent with hypotheses, to the extent that participants trusted and consumed Fox News, they perceived lower election legitimacy (main effect of Fox News ratings: $b$ = –0.20), that decreased over time (election timing × Fox News ratings: $b$ = –0.17). To the extent participants trusted and consumed other outlets, they perceived higher election legitimacy (main effect of other outlet consumption: $b$ = 0.40) that increased over time (election timing × other outlets ratings: $b$ = 0.28). Consumption and trust of polarized media predicted changes over time in perceptions of election legitimacy.

Moreover, the analysis yielded a 4-way interaction (partisan identification × election timing × other outlets ratings × Fox News ratings: $b$ = –0.29). We decomposed the interaction by partisan identification. Among Democrats, trust and consumption of Fox News, independent of ratings of other outlets, was associated with reductions in the increase of perceptions of

**Table 1. Multiple regression predicting perceived election legitimacy from participant partisan identification, election timing, and their interaction (Model 1) and additionally from the mean centered averages of perception of Fox News and of other media outlets, and their interactions (Model 2).**

| Predictor | Model 1 | | | | Model 2 | | | |
|---|---|---|---|---|---|---|---|---|
| | *b* | *SE* | *t* | *p* | *b* | *SE* | *t* | *p* |
| **Partisan Identification** | | | | | | | | |
| Democrat vs. Republican | −1.26 | 0.07 | −17.66 | < .001 | −0.97 | 0.09 | −10.46 | < .001 |
| Independent vs. Democrat and Republican | 0.55 | 0.09 | 6.11 | < .001 | 0.54 | 0.10 | 5.28 | < .001 |
| **Election Timing** | | | | | | | | |
| Election timing | 0.08 | 0.07 | 1.16 | .248 | 0.15 | 0.09 | 1.69 | .091 |
| **Media Measures** | | | | | | | | |
| Fox News trust and consumption | – | – | – | – | −0.20 | 0.04 | −5.07 | < .001 |
| Other media trust and consumption | – | – | – | – | 0.40 | 0.05 | 7.66 | < .001 |
| **Partisan Identification x Election Timing Interactions** | | | | | | | | |
| Democrat vs. Republican x Election timing | −0.65 | 0.14 | −4.58 | < .001 | −0.27 | 0.19 | −1.48 | .140 |
| Independent vs. Democrat and Republican x Election timing | 0.10 | 0.18 | 0.56 | .574 | 0.11 | 0.21 | 0.51 | .607 |
| **Media Measures x Election Timing Interactions** | | | | | | | | |
| Fox News trust and consumption x Election timing | – | – | – | – | −0.17 | 0.08 | −2.20 | .028 |
| Other media trust and consumption x Election timing | – | – | – | – | 0.28 | 0.10 | 2.71 | .007 |
| **Partisan Identification x Media Measures x Election Timing Interactions** | | | | | | | | |
| Democrat vs. Republican x Election timing x Fox trust and consumption | – | – | – | – | 0.22 | 0.14 | 1.61 | .109 |
| Independent vs. Democrat and Republican x Election timing x Fox trust and consumption | – | – | – | – | −0.02 | 0.20 | −0.11 | .915 |
| Democrat vs. Republican x Election timing x Other media trust and consumption | – | – | – | – | 0.59 | 0.22 | 2.62 | .009 |
| Independent vs. Democrat and Republican x Election timing x Other media trust and consumption | – | – | – | – | −0.09 | 0.24 | −0.38 | .704 |
| **Partisan Identification x Both Media Measures x Election Timing Interaction** | | | | | | | | |
| Democrat vs. Republican x Election timing x Fox trust and consumption x Other media trust and consumption | – | – | – | – | −0.29 | 0.14 | −2.10 | .036 |
| Independent vs. Democrat and Republican x Election timing x Fox trust and consumption x Other media trust and consumption | – | – | – | – | −0.20 | 0.17 | −1.21 | .226 |

*Note*: Model 1: $R^2$ = 0.248, df = 1202; Model 2: $R^2$ = 0.312, df = 1184. Regressions included two contrast coded predictors: One compared Democrats (−1/2) with Republicans (+1/2), and one compared Independents (−2/3) with Democrats and Republicans (+1/3 for both).

election legitimacy over time (election timing × Fox News ratings: $b$ = −0.29, $t$ = −3.04, $p$ = 0.002). Democratic ratings of other outlets were not associated with changes over time in perceived election legitimacy (election timing × Other Outlet ratings: $b$ = −0.04, $t$ = −0.42, $p$ = .676). Among Republicans, in contrast, trust and consumption of other media outlets, independent of Fox News ratings, was associated with a reduced decline in perceived election legitimacy (election timing × Other Outlet ratings: $b$ = 0.54, $t$ = 2.89, $p$ = .040). Higher Republican ratings of Fox News were not associated with changes over time in perceived election legitimacy (election timing × Fox News ratings: $b$ = −0.07, $t$ = −0.74, $p$ = .458).

Thus, to the extent that participants trusted and consumed news outlets that are typically distrusted and avoided by their political in-group—Fox News for Democrats and other sources for Republicans—their perceptions of election legitimacy followed the prevailing pattern within their political group less strongly. We did not, however, find that trusting and consuming news outlets that are typically aligned with political in-group—other outlets for Democrats and Fox News for Republicans—were associated with stronger prevailing patterns within their in-group. We are hesitant to draw overly strong conclusions based on an exploratory analysis of higher order interactions. Yet the results are consistent with the possibility that trust and consumption of less ideologically diverse media outlets is associated with greater polarized perceptions of election legitimacy.

## Perceived election legitimacy of own vote and nationwide vote

Past findings suggest that people trust their local government more than they trust the national government [45]. This raises the possibility that people who lose elections lose faith in the

national election legitimacy because they suspect that other people's (i.e., those from other areas) votes are not counted as intended. In the perceived election legitimacy analysis reported above, there was a significant 3-way interaction between partisan identification, timing, and vote type ($F(2, 1201) = 8.71$, $p < .001$, $\eta_p^2 = .014$). The partisanship × timing interaction was larger for perceived legitimacy of nationwide ($F(2, 1201) = 17.89$, $p < .001$, $\eta_p^2 = .029$) than for own votes ($F(2, 1202) = 3.60$, $p = .027$, $\eta_p^2 = .006$). From the Undeclared to Declared period, Democrats' perceived legitimacy increased for both nationwide votes ($M_{\text{Undeclared}} = 3.78$, $SD_{\text{Undeclared}} = 1.10$, $M_{\text{Declared}} = 4.36$, $SD_{\text{Declared}} = 0.94$; $F(1, 1201) = 33.18$, $p < .001$, $\eta_p^2 = .027$) and own votes ($M_{\text{Undeclared}} = 4.07$, $SD_{\text{Undeclared}} = 1.03$, $M_{\text{Declared}} = 4.37$, $SD_{\text{Declared}} = 0.97$; $F(1, 1202) = 8.01$, $p = .005$, $\eta_p^2 = .007$). Among Republicans, from the Undeclared to the Declared period, the decrease in perceived legitimacy of nationwide votes was larger ($M_{\text{Undeclared}} = 2.64$, $SD_{\text{Undeclared}} = 1.25$, $M_{\text{Declared}} = 2.33$, $SD_{\text{Declared}} = 1.33$; $F(1, 1201) = 7.56$, $p = .006$, $\eta_p^2 = .006$) than for own votes, which did not significantly decrease ($M_{\text{Undeclared}} = 3.35$, $SD_{\text{Undeclared}} = 1.34$, $M_{\text{Declared}} = 3.24$, $SD_{\text{Declared}} = 1.44$; $F(1, 1202) = 0.93$, $p = .335$, $\eta_p^2 = .001$). The differential decline for Republicans may reflect that people know more about their own votes than they do about others' votes. For their own votes, people have direct, personal experience voting and trust in their local district's counting process whereas the uncertainty and lack of experience with nationwide votes may lend itself more to rationalizing processes. An analogous difference among Democrats' between confidence in one's own vote versus nationwide votes may not have emerged because of a ceiling effect for confidence.

## Discussion

Citizens in healthy democracies view votes as legitimate even when their preferred candidate loses [2–4]. Judging elections as fair and legitimate is critical for governments to operate effectively [4, 43]. In contrast with these democratic ideals, election losers tend to perceive elections as less legitimate than do winners [10, 18]. Researchers have attributed such outcome-dependent perceptions of election legitimacy to the reduction of cognitive dissonance wherein losers perceive elections as less legitimate, with the implication that in a legitimate election their preferred candidate would have won more votes [18]. Consistent with this reasoning, polarized perceptions of election legitimacy increase from pre- to post-election. In previous research, however, knowledge of the outcome from pre- to post-election was confounded with the possibility of voting or influencing votes, which is itself sufficient to polarize election attitudes. The present study used a naturalistic experiment to demonstrate that learning the election outcome increases polarized perceptions of election legitimacy, even when it is no longer possible to exert electoral influence via voting, consistent with theories of cognitive dissonance.

In the context of the U.S. 2020 presidential election, Democrats (winners) were more confident than were Republicans (losers) that both their own and nationwide votes were counted correctly. These polarized perceptions increased from Election Day through the second week following Election Day as evidence accumulated that President Joe Biden won. This design removes the possibility of influencing votes, more precisely implicating knowledge of the election outcome in provoking dissonance. As more votes were counted and Democrats' expected Biden win was confirmed, they became more confident that the votes were counted correctly. Over the same period, as more votes were counted and Republicans' expected Trump win was disconfirmed, they became less confident that votes were counted correctly. Along with perceptions of election legitimacy, partisans' emotions became increasingly polarized. From the undeclared to the declared period, Democrats' emotions became more prevailingly positive and less negative while Republicans' emotions became less positive and more prevailingly

negative. These emotional profiles may both reflect the arousal of dissonance [22–24] and increasingly confident assessments that the election was illegitimate or legitimate.

These polarized assessments of electoral integrity and emotions corresponded with polarized media. Trust and consumption of Fox News independently predicted lower perceived election legitimacy that decreased over time. Trust and consumption of other media outlets independently predicted higher perceived election legitimacy that increased over time. These findings are consistent with the possibility that polarized media functions as social confirmation that polarizes perceptions of election legitimacy [26, 28, 38, 44].

The present findings illustrate a point made by Eliot Aronson and Carol Tavris [45], that social confirmation by groups with whom one is strongly connected facilitates dissonance-reducing rationalizations that can distort evidence, or as they put it: "[W]hen people feel a strong connection to a political party, leader, ideology, or belief, they are more likely to let that allegiance do their thinking for them and distort or ignore the evidence that challenges those loyalties" [45].

It is noteworthy that essays by both Lee Ross [8] and Eliot Aronson [45], two prominent social psychologists reflecting on the relevance of cognitive dissonance theory to contemporary issues, focus on role of the collective in fomenting rationalization. Although this was not the focus of much cognitive dissonance research, which emphasized intraindividual rather than collective processes, their observations resonate with one of the classic studies conducted by Festinger and colleagues. In their study of the reactions of a doomsday cult whose prophesied alien arrival failed to materialize, Festinger and colleagues explained how social processes enable belief persistence when confronted with disconfirming evidence [7, p. 4]:

> The individual believer must have social support. It is unlikely that one isolated believer could withstand the kind of disconfirming evidence we have specified. If, however, the believer is a member of a group of convinced persons who can support one another, we would expect the belief to be maintained. . .

In other words, reducing cognitive dissonance to maintain disconfirmed expectations is not simply an individual rationalization process, but is bolstered by social support, justifications, and the provision of rationalizing information by others. Although media outlets are different from tightly knit social groups like cults, media may serve a similar function by providing polarized information that helps protect and confirm beliefs shaken by disconfirming evidence, processes hinted at by our findings regarding trust and consumption of Fox News and other mainstream media.

Fox News was uniquely trusted and consumed by Republicans and not Democrats, but this deserves further comment. As noted earlier, our selection of media outlets reflected the availability of searchable full text databases of media content for use in a different project. We also suspected that the Wall Street Journal would be trusted and consumed by Republicans more than Democrats, and that outlets like USA Today and traditional network television would be rated equally by Democrats and Republicans. But this was not the case. Right-wing media outlets have increased in recent years, as Republicans became less trusting of so-called "mainstream media," which President Trump had referred to as "fake news" [43, 44]. An important task for future research will be to study the number of outlets trusted and consumed by Republicans.

Future research might also pursue several additional clarifying questions. One would be to differentiate the influence of polarized media from that of political leaders such as President Joe Biden and President Donald Trump. Another would be to investigate why people have more confidence that their own votes were counted correctly than that nationwide votes were

counted correctly, much as they trust their local government more than they trust the national government [46]. People may have more direct knowledge that their ballot was clearly completed and mailed or deposited in the ballot box, whereas their knowledge of nationwide ballots are indirect, distant, and open to suspicion or reinterpretation. Examining an expanded time course of polarized perceptions of election legitimacy is another important question for future work. The present study was limited to the two weeks following Election Day so cannot directly address questions about the persistence of polarized perceptions, as would be implied by cognitive dissonance theory. However, evidence from national polls indicate that polarization persisted until late January 2021, nearly three months after the election [47]. And, of course, the January 6, 2021, attack on the U.S. Capitol suggests that persistent assessment of an illegitimate election was sufficient to motivate acts of political violence [48].

Notwithstanding these open questions, the present findings have implications for understanding and improving media engagement to reduce polarized perceptions of election legitimacy. Others have noted how Fox News undermines perceptions of election legitimacy [37, 40]. The present results additionally undergird the importance of consuming ideologically diverse media sources to mitigate polarized perceptions of election legitimacy, a theme worth emphasizing in civic education.

Pursuing these topics is important because they inform understanding of both psychology and democracy. A recent analysis found that perceived electoral integrity is less conditional on electoral outcomes in healthy democracies. Healthier democracies not only evince less of a negative effect on perceived electoral integrity among the losing party but also less of an increase among the wining party [49]. In the U.S. 2020 presidential election, as evidence of an election's outcome accumulated, vote confidence was conditional on knowledge of the election outcome in both directions—winners became more confident that votes were correctly counted whereas losers became less confident. Social confirmation abets the dissonance reducing process for winners and losers alike, illustrating how politically polarized vote confidence is both fomented and reflected by polarized media information ecosystems. And cognitive dissonance theory, when explored in the context of contemporary events of dramatic consequence, continues to yield insights and inspire new research questions.

## Acknowledgments

Lee Ross passed away during the writing of this manuscript, and the posthumous essay on cognitive dissonance theory cited in this paper emerged during the celebration of Ross's contributions to social psychology in the wake of his passing. Interested readers can find this and other essays in the forthcoming book [8].

## Author Contributions

**Conceptualization:** Marrissa D. Grant, Alexandra Flores, David K. Sherman, Leaf Van Boven.

**Data curation:** Marrissa D. Grant, Alexandra Flores.

**Formal analysis:** Marrissa D. Grant, Alexandra Flores, Eric J. Pedersen, Leaf Van Boven.

**Methodology:** Marrissa D. Grant, Alexandra Flores, David K. Sherman, Leaf Van Boven.

**Project administration:** Alexandra Flores, Leaf Van Boven.

**Supervision:** Leaf Van Boven.

**Writing – original draft:** Marrissa D. Grant, Alexandra Flores, David K. Sherman, Leaf Van Boven.

**Writing – review & editing:** Marrissa D. Grant, Alexandra Flores, Eric J. Pedersen, David K. Sherman, Leaf Van Boven.

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
