## [Decision Letter · Decision Letter 0]

10 Aug 2021

PONE-D-21-23399

When Election Expectations Fail: 

Polarized Perceptions of Election Legitimacy Increase with Accumulating Evidence of Election Outcomes and with Polarized Media

PLOS ONE

Dear Dr. Van Boven,

Thank you for submitting your manuscript to PLOS ONE. After careful consideration, we feel that it has merit but does not fully meet PLOS ONE’s publication criteria as it currently stands. Therefore, we invite you to submit a revised version of the manuscript that addresses the points raised during the review process.

I asked two experts in the field to review this manuscript. I synthesize their key comments with my own (based on an independent reading) below. Overall, all three of us think that this paper should be published in PLOS ONE, though we differ somewhat in how much more work is needed.

The two places that I think need to be revised are the positioning and (at least one of) the analyses. For the positioning, you lean heavily on dissonance theory, which certainly makes sense intuitively, and yet as R2 points out, there are other plausible explanations for why belief in the election’s legitimacy could change with time. Beyond that, there is no direct measure of dissonance, making it that much more difficult to know if dissonance is driving this result or not. I would suggest significantly reducing the exposition on dissonance and alluding to the entire idea as just a possibility, rather than a theory to build on (with these results being confirmation of that theory). For the analyses, I agree with R2’s belief that the polarization effect should (based on your set up) be moderated by in-group media consumption. I encourage you to run the set of analyses that the reviewer suggested. If the results don't support your theorizing, I would suggest that you both downplay the media consumption angle entirely AND report the lack of a result in an appendix.

Aside from these two larger issues, I suggest you carefully read and respond to the other comments made by both reviewers as they clearly took the care to provide feedback that would improve your work.

On the whole, I think that with these changes, this manuscript has a clear path to publication. Short of something significantly new in the revision, I do not anticipate sending the manuscript back out to review.

As always, please provide a detailed summary of the changes you have made in response to the entire review team’s comments.

Best of luck.

We look forward to receiving your revised manuscript.

Kind regards,

Jeff Galak, PhD

Academic Editor

PLOS ONE

Journal Requirements:

Reviewers' comments:

Reviewer's Responses to Questions

**Comments to the Author**

1. Is the manuscript technically sound, and do the data support the conclusions?

Reviewer #1: Yes

Reviewer #2: Yes

2. Has the statistical analysis been performed appropriately and rigorously? 

Reviewer #1: I Don't Know

Reviewer #2: Yes

3. Have the authors made all data underlying the findings in their manuscript fully available?

Reviewer #1: Yes

Reviewer #2: Yes

4. Is the manuscript presented in an intelligible fashion and written in standard English?

Reviewer #1: Yes

Reviewer #2: Yes

5. Review Comments to the Author

Reviewer #1: I was a reviewer for this manuscript when it was previously submitted to a different journal, and I very much appreciate how the authors have altered the paper to reflect and address many of my earlier concerns. I also thought that the new material from Lee Ross nicely motivated the present investigation. What follows are some lingering issues that I think can easily be addressed in a revision:

Line 82: Unfortunately, I can’t quite parse the following sentence: “For winners, confirming knowledge that their expected and hoped-for win needs to be psychologically reconciled with any lingering doubts about the candidate’s electability and other perceptions of the candidate’s weaknesses” -- even by manipulating whether “confirming” is intended as a verb or an adjective.

Line 102: “The present study also provides evidence for the role of social confirmation in reducing dissonance through motivated reasoning about election legitimacy” – again a bit difficult to parse -- is the role played by social confirmation or motivated reasoning? I realize that “dissonance through motivated reasoning about election legitimacy” is intended to be a single unit, but maybe unpacking the sentence a bit would be helpful.

Line 119: I appreciate the new edit, but “high” should be “highly,” yes?

Line 243: I also appreciate this acknowledgement but maybe it’s also worth acknowledging somewhere that, especially in light of the handy figure, the outcome of the election was somewhat of a continuing process, not a bright-line distinction.

Line 298-303: Democrats’ conviction that their votes were counted correctly evidently increased much more than the barely significant drop in Republicans’ confidence that their votes were counted correctly, yes? Maybe worth mentioning that difference?

More importantly, why do these means differ from those reported in the next paragraph in

Lines 312-313 and Lines 317-318? No doubt I have missed some distinction, but I’d nevertheless appreciate a clarification.

Line 318-320: “This differential decline may simply reflect that people know more about their own votes than they do about others’ votes.” Perhaps, but then why don’t we see a similar increase for Democrats’ confidence in their own vote vs. others’ votes? Perhaps a ceiling effect?

Line 437: There appears to be a word missing

Line 438: Given the measures employed in the study, it’s a bit odd to refer to, for example, Fox News as a source of collective dissonance reduction in the same way that a doomsday cult (or other tightly-knit social groups) apparently was. The latter obviously offered a much more intimate connection for the target perceiver than the former, which could simply (or primarily) represent a source of biased information, as opposed to all the other ways in which fellow cult members could be a source of comfort when a prophecy is seemingly disconfirmed. Perhaps some such acknowledgment of this distinction is warranted.

Reviewer #2: I have to admit that I have never really understood PLOS ONE’s publication criteria and the role of PLOS ONE reviewers. If my task was to examine if the manuscript reports empirical research that satisfies the basic standards of science and if the description of the research is comprehensible for scientists from other fields and practitioners then my answer is “Yes this manuscript should be published in PLOS ONE.” In other words, the manuscript satisfies the 7 criteria listed in the Guidelines for Reviewers.

Below a more nuanced review that highlights the strengths and weaknesses of the manuscript. I am providing rather detailed input to give the authors the opportunity to increase the scientific contribution and impact of their paper.

There are clearly many things to like about this manuscript. The experimental manipulation is a strong point. So is the theoretical research question as well as the attempt to test theoretical predictions about cognitive dissonance in an applied setting. The study is well-conducted, and the analyses are mostly correct and reported in a straightforward manner. I liked the violin density plots. The findings provide evidence for the idea that emotions and perceptions of election legitimacy polarized over time after the 2020 presidential election.

Neither the predictions nor the results are particularly surprising. Most people, even those without any training in social sciences, would predict that losers perceive elections as less legitimate (and experience more negative emotions) than winners, and that this difference increases as the outcome of the elections become more and more clear. Once does not need cognitive dissonance to explain this effect.

The difference between the two groups, “Declared” and “Undeclared,” is not just that Biden was declared the winner of the elections. Many other things happened between Nov. 4 and 15, 2020. Trump and many Republican elected officials declared the elections as being rigged, whereas Biden and his team kept saying that the elections were legitimate. The observed results could be due to cognitive dissonance reduction, as the authors claim. But they could also be due to respondents simply being influenced by their party leaders or numerous other things that changed between Nov. 4 and 15. Although the study contains an experimental manipulation (= is a “true experiment”) it is unclear what was manipulated here.

Given the interpretational ambiguities of the results, I feel that the authors overstate their results. I think the authors should adopt more cautious language and delete sentences such as “These findings advance theoretical understanding of polarized perceptions of election legitimacy by more directly implicating rationalizing processes associated with cognitive dissonance and motivated reasoning” and “[…] suggest the role of emotion in the arousal and reduction of dissonance through rationalization.”

Has this study been preregistered? I am asking because the authors made many choices that I would not have made. Here some examples. Party affiliation was measured on a 7-point scale. The authors trichotomized this continuous scale into three categories: Responses 1, 2, and 3 were labeled “Democrat,” response 4 “Independent,” and 4, 6, and 7 “Republican.” Why not a different categorization into responses 1 and 2 (Democrat), 3, 4, and 5 (Independent), and 6 and 7 (Republican)? Or why not treat this variable as a continuous predictor with 7 levels? The latter choice would have made sense because the “quadratic trend” is significant (see Table 1). Where does the idea come from to combine all “other” 14 media outlets into one score which is then contrasted to Fox News?

Given the introduction, I expected the polarizing effect (the increase in difference between Democrats and Republicans between Undeclared and Declared) to be moderated by in-group media consumption. In other words, I expected the authors to test a 3-way interaction between party identification, timing, and media consumption (the latter being a continuous predictor indicating the extent to which respondent consume media known to promote ideas consistent with the respondents’ party identification). I also expected the authors to test a moderated mediation model in which the three-way interactive effect on election legitimacy is moderated by emotions, i.e., the difference between negative and positive emptions. Such a moderated mediation model corresponds to Hayes’ Model #11.

Minor points:

I was confused by the fact that different terms were used interchangeably. “Party identification” was sometimes called “party identity” and “partisan ID.” “Election legitimacy” was sometimes referred to as “vote legitimacy,” “perception of legitimacy,” and “confidence in vote legitimacy.” “Media outlets” are also “news outlets” and “sources.” “Media trust and consumption” is also called “engagement.” “National votes” are sometimes “nationwide votes.”

I didn’t understand the sentence “Respondents answered both questions on non-numeric scales” (p. 13). Are the 5-point scales, which were mentioned two sentences before, non-numeric? A similar issue occurs on page 13 where the authors say “Participants answered on two scales presented without numbers (1 = Not at all confident; 5 = Very confident).” Do the authors mean to say that they presented respondents with five verbal labels and that they later assigned the numbers 1 to 5 to these labels?

The choice of the 15 media outlets is surprising. How come the authors did not include a larger number of conservative outlets?

It is not very informative to compute for each respondent a correlation between trust and consumption across the 15 media outlets, i.e., 1236 correlations each with an N of 15 (p. 13-14). It would be better to compute for each media outlet a correlation across all participants, i.e., 15 correlations each with an N of 1236. The authors should then report the median and the range of these 15 correlations.

On page 14, the authors report a one-df test, F(1, 1078) = 3.13, p = .077, which is a 2-df test.

I’d drop the factor “vote type” in the analyses reported on pages 14-15 and I’d collapse the top and bottom panels in Figure 2). The factor doesn’t add anything to the paper. Given that the two vote confidence ratings are averaged in the remaining analyses, including it as a factor in the earlier analyses creates confusion for readers. The fact that the party identification by timing interaction was stronger for national votes can be mentioned in a footnote.

I did not understand right away what the authors meant by “systematic legitimacy of national votes” (p. 15).

It says on page 15 that “Republicans’ confidence that their own votes were correctly counted did not significantly decrease over time (M_Undeclared= 3.31, […] M_Declared = 3.31)”, but then the red line in the top middle panel in Figure 2 is not perfectly flat. How is this possible?

Figure 2: I’d put the violin density plots for the Independents in the middle rather than on the right side.

The authors say on page 17 “In a regression analysis, negative emotions predicted lower confidence that votes were counted correctly,” but it is not clear what confidence ratings they are referring to, the “own vote,” the “nationwide vote,” or the average of the two ratings. The same issue exists in the title of Table 1 and the analyses reported on p. 20.

It is incorrect to say that Figure 4 is a graphic representation of the result that “Democrats’ and Republicans’ differential engagement with polarized media corresponded with their polarized confidence in vote legitimacy” (p. 19). Confidence in vote legitimacy (= perceived election legitimacy) is not shown in Figure 4.

The significant “curvilinear trend” (b = 0.16) reported in Table 1 is surprising. This coefficient and Figure 2 suggest that the polarizing effect is mostly due to Democrats becoming more confident. Is this finding consistent with the authors’ theoretical analyses? Wouldn’t we expect cognitive dissonance effects be strongest for losers?

I don’t understand why the authors first include media outlet (= source) as a within-subject factor (which is identical to computing a difference score; see p. 18) but then run analyses in which they include both media outlet scores as predictors (Table 1). Is type of media outlet hypothesized to be a moderator, or are the authors predicting the existence of two (parallel?) mediators?

The sentence “It is noteworthy how in essays …” (p. 22) is formulated in an awkward manner.

The authors dedicate four paragraphs to the types of questions that future research might examine (pages 23-25). I think these ideas can be reduced to one paragraph. It would be more interesting for the authors to discuss the implications of their findings rather than provide a list of the numerous things they didn’t do in the present research.

6. PLOS authors have the option to publish the peer review history of their article (what does this mean?). If published, this will include your full peer review and any attached files.

Reviewer #1: No

Reviewer #2: No

---

## [Author Response · Author response to Decision Letter 0]

15 Oct 2021

PONE-D-21-23399

When Election Expectations Fail: Polarized Perceptions of Election Legitimacy Increase with Accumulating Evidence of Election Outcomes and with Polarized Media

Friday, October 15, 2021

Dear Professor Galek:

This letter accompanies the resubmission of PONE-D-21-23399, now titled, “When Election Expectations Fail: Polarized Perceptions of Election Legitimacy Increase with Accumulating Evidence of Election Outcomes and with Polarized Media.” The manuscript is co-authored by me, Marrissa “Dani” Grant, Alexandra Flores, Eric Pedersen, and David Sherman. Note that we have reordered authorship (Grant, Flores, Pedersen, Sherman, & Van Boven) to better reflect the contributions of the two lead authors. 

Thank you and the reviewers for the thoughtful and incisive comments. The quality of reviews was of the highest caliber. We have revised the manuscript in response to reviewer comments in ways that, we believe, led to a manuscript with clearer and more substantial contributions. Below, we provide a summary of three larger revisions, followed by detailed responses to your and the reviewers’ comments. 

(1) You requested that we conduct the analysis suggested by Reviewer 2, specifically, “to test a 3-way interaction between party identification, timing, and media consumption (the latter being a continuous predictor indicating the extent to which respondent consume media known to promote ideas consistent with the respondents’ party identification).” We now report these analyses in Table 1 and in the Results section (lines 369-382). 

The findings are consistent with moderation implied by our hypotheses. To the extent that participants trusted and consumed Fox News, they perceived lower election legitimacy (main effect of Fox News ratings: b = –0.20), that decreased over time (election timing × Fox News ratings: b = –0.17). To the extent participants trusted and consumed other outlets, they perceived higher election legitimacy (main effect of other outlet consumption: b = 0.40) that increased over time (election timing × other outlets ratings: b = 0.28). These findings indicate that trust and consumption of Fox News and other outlets differentially predict perceptions of vote legitimacy. 

Moreover, the analysis yielded a 4-way interaction (partisan identification × election timing × other outlets ratings × Fox News ratings: b = –0.29). For Democrats, higher ratings of Fox News was associated with reductions in the increase of perceptions of election legitimacy over time (election timing × Fox News ratings: b = –0.29, t = –3.04, p = .002). Higher Democratic ratings of other outlets were not associated with changes over time in perceived election legitimacy (election timing × Other Outlet ratings: b = –0.04, t = –0.42, p = .676). Among Republicans, in contrast, trust and consumption of other media outlets was associated with a reduced decline in perceived election legitimacy (election timing × Other Outlet ratings: b = 0.54, t = 2.89, p = .040). Higher Republican ratings of Fox News were not associated with changes over time in perceived election legitimacy (election timing × Fox News ratings: b = –0.07, t = –0.74, p = .458). Thus, to the extent that participants trusted and consumed news outlets that are typically distrusted and avoided by their political in-group—Fox News for Democrats and other sources for Republicans—their perceptions of election legitimacy less strongly followed the prevailing pattern within their political group. 

As we state in the manuscript, however, we are hesitant to draw overly strong conclusions from an exploratory analysis of higher order interactions. We chose not to explore more complicated moderated mediation and mediated moderation analyses given the potentially complicating presence of a 4-way interaction, the exploratory nature of mediation analyses, and our general hesitancy about unwarranted causal inferences given correlational mediation analyses. 

(2) You asked us to scale back reliance on cognitive dissonance theory, depending on the outcome of requested analyses. As noted above, the moderation analysis is generally consistent with hypotheses the the degree to which people trust and consume polarized media sources, they exhibit polarized perceptions of election legitimacy that increase over time. These findings do not allow strong causal conclusions, however. We have adopted more cautious language throughout the paper, clarifying that although we derive predictions from dissonance theory and although our study provides a more precise test of those predictions than previous research on perceptions of election legitimacy, the findings are nevertheless open to alternative interpretations and cannot be exclusively connected to cognitive dissonance theory. 

(3) We have revised our theoretical and empirical analyses of emotion. You and the reviewers noted that negative emotion is not a precise measure of dissonance, and that it is difficult to know whether emotions are driving or are driven by polarized perceptions of election legitimacy. We agree and acknowledge that our previous discussion of emotion was overly narrow. Clearly, negative emotions are as much a consequence of perceptions that the election was illegitimate as they are a driver of perceived election legitimacy. Indeed, in an exploratory serial mediation analysis, there was a significant indirect effect from partisan identity to media perception to vote confidence to relative emotion, reflecting that people felt more negative (and less positive) to the extent they were skeptical that votes were counted correctly. Given the reciprocal relationship between emotion and perceived election legitimacy, we report the effects of partisan identity and timing on emotion, and we report the correlations between emotion and perceived election legitimacy (controlling for identity and timing), but we do not make or test causal claims. We have, accordingly, revised our theoretical analysis of emotion in the introduction and elsewhere. 

In addition to these three broader revisions, we have made relatively minor revisions in response to comments and suggestions, as detailed below. 

EDITOR COMMENTS

Dear Dr. Van Boven,

Thank you for submitting your manuscript to PLOS ONE. After careful consideration, we feel that it has merit but does not fully meet PLOS ONE’s publication criteria as it currently stands. Therefore, we invite you to submit a revised version of the manuscript that addresses the points raised during the review process.

I asked two experts in the field to review this manuscript. I synthesize their key comments with my own (based on an independent reading) below. Overall, all three of us think that this paper should be published in PLOS ONE, though we differ somewhat in how much more work is needed.

The two places that I think need to be revised are the positioning and (at least one of) the analyses. For the positioning, you lean heavily on dissonance theory, which certainly makes sense intuitively, and yet as R2 points out, there are other plausible explanations for why belief in the election’s legitimacy could change with time. Beyond that, there is no direct measure of dissonance, making it that much more difficult to know if dissonance is driving this result or not. I would suggest significantly reducing the exposition on dissonance and alluding to the entire idea as just a possibility, rather than a theory to build on (with these results being confirmation of that theory). 

See our point (2) above. We have clarified that we derive predictions from cognitive dissonance theory, while acknowledging that our findings may also be consistent with other interpretations. As detailed below, however, part of Reviewer 2’s alternative interpretation, that different media outlets provided different information during the week following election day, is the same explanation implied by cognitive dissonance theory, as articulated in Festinger and colleagues’ “When Prophecy Fails….” 

For the analyses, I agree with R2’s belief that the polarization effect should (based on your set up) be moderated by in-group media consumption. I encourage you to run the set of analyses that the reviewer suggested. If the results don't support your theorizing, I would suggest that you both downplay the media consumption angle entirely AND report the lack of a result in an appendix.

As discussed in point (1) above, we conducted the exploratory moderation analysis (lines 369-382). 

As discussed in point (3), we acknowledge the reciprocal role of emotion and legitimacy. We report the effects of partisanship and timing on emotion and the correlation between emotion and perceived election legitimacy, controlling for partisanship and timing, but do not report mediation analyses (lines 331-358). 

Aside from these two larger issues, I suggest you carefully read and respond to the other comments made by both reviewers as they clearly took the care to provide feedback that would improve your work.

We have addressed the reviewers’ comments, as specified in the following rebuttal letter. 

On the whole, I think that with these changes, this manuscript has a clear path to publication. Short of something significantly new in the revision, I do not anticipate sending the manuscript back out to review. As always, please provide a detailed summary of the changes you have made in response to the entire review team’s comments.

Best of luck.

Thank you for the encouragement. We look forward to learning your decision. 

REVIEW COMMENTS TO THE AUTHOR

Reviewer #1: I was a reviewer for this manuscript when it was previously submitted to a different journal, and I very much appreciate how the authors have altered the paper to reflect and address many of my earlier concerns. I also thought that the new material from Lee Ross nicely motivated the present investigation. What follows are some lingering issues that I think can easily be addressed in a revision:

Line 82: Unfortunately, I can’t quite parse the following sentence: “For winners, confirming knowledge that their expected and hoped-for win needs to be psychologically reconciled with any lingering doubts about the candidate’s electability and other perceptions of the candidate’s weaknesses” -- even by manipulating whether “confirming” is intended as a verb or an adjective.

Thank you for pointing out the confusing wording. We have revised for clarity (lines 80-82).

Line 102: “The present study also provides evidence for the role of social confirmation in reducing dissonance through motivated reasoning about election legitimacy” – again a bit difficult to parse -- is the role played by social confirmation or motivated reasoning? I realize that “dissonance through motivated reasoning about election legitimacy” is intended to be a single unit, but maybe unpacking the sentence a bit would be helpful.

Thanks for catching this, we have revised this sentence for clarity (lines 96-98).

Line 119: I appreciate the new edit, but “high” should be “highly,” yes?

Yes. We have clarified (line 115).

Line 243: I also appreciate this acknowledgement but maybe it’s also worth acknowledging somewhere that, especially in light of the handy figure, the outcome of the election was somewhat of a continuing process, not a bright-line distinction.

Thank you for pointing out this ambiguity, which we have clarified (lines 195-198).

Line 298-303: Democrats’ conviction that their votes were counted correctly evidently increased much more than the barely significant drop in Republicans’ confidence that their votes were counted correctly, yes? Maybe worth mentioning that difference?

Yes, the increase in Democrats’ confidence that votes were counted correctly was larger in magnitude (MUndeclared = 3.92, SD = 1.00; MDeclared = 4.37, SD = 0.92; F(1, 1201) = 20.98, p < .0001) than the decrease in Republicans’ confidence in vote counts (MUndeclared = 2.99, SD = 1.18; MDeclared = 2.79, SD = 1.24; F(1, 1201) = 3.95, p = .047). We acknowledge this difference (lines 319-323) and speculate that it may simply reflect that Democrats’ increase in vote legitimacy aligned with accumulating evidence that votes were legitimately counted. For Republicans, in contrast, their decreasing confidence in vote counts occurred despite accumulating evidence that votes were correctly counted. 

More importantly, why do these means differ from those reported in the next paragraph in Lines 312-313 and Lines 317-318? No doubt I have missed some distinction, but I’d nevertheless appreciate a clarification.

Thank you for pointing out this unclear description. We have chosen to separate these paragraphs into two sections to demonstrate the exploratory nature of the test that includes own vote confidence and nationwide vote confidence separately. In the first section, we report the means and analyses for a perceived election legitimacy which is the average of the own vote confidence and nationwide vote confidence ratings (lines 261-262 for definition of measure, lines 311-318 for results). In an exploratory analysis at the end of the results section, we report separately the means for confidence that own and nationwide votes were correctly counted (lines 431-447). We have also corrected minor errors in reporting means (owing to inadvertent exclusion of people who completed the survey on November 8), which have been corrected.

Line 318-320: “This differential decline may simply reflect that people know more about their own votes than they do about others’ votes.” Perhaps, but then why don’t we see a similar increase for Democrats’ confidence in their own vote vs. others’ votes? Perhaps a ceiling effect?

This is a good point. We have included an acknowledgement that a ceiling effect might have restricted Democrats’ increase in confidence that their own votes were counted correctly (lines 447-449).

Line 437: There appears to be a word missing

We have clarified the sentence (line 491-494).

Line 438: Given the measures employed in the study, it’s a bit odd to refer to, for example, Fox News as a source of collective dissonance reduction in the same way that a doomsday cult (or other tightly-knit social groups) apparently was. The latter obviously offered a much more intimate connection for the target perceiver than the former, which could simply (or primarily) represent a source of biased information, as opposed to all the other ways in which fellow cult members could be a source of comfort when a prophecy is seemingly disconfirmed. Perhaps some such acknowledgment of this distinction is warranted.

We acknowledge the differences more clearly between media outlets and tightly knit social groups (lines 503-507), while speculating that the informational function, if not the social connection functions, of media sources may be similar to social groups. 

~~~~~~~~~~~~~~~~~~~~~~~~~~~~~~~~~~~~~~~~~~~~~~~~~~~~~~~~~~~~~~~~~~~~~~~~

Reviewer #2: I have to admit that I have never really understood PLOS ONE’s publication criteria and the role of PLOS ONE reviewers. If my task was to examine if the manuscript reports empirical research that satisfies the basic standards of science and if the description of the research is comprehensible for scientists from other fields and practitioners then my answer is “Yes this manuscript should be published in PLOS ONE.” In other words, the manuscript satisfies the 7 criteria listed in the Guidelines for Reviewers.

Below a more nuanced review that highlights the strengths and weaknesses of the manuscript. I am providing rather detailed input to give the authors the opportunity to increase the scientific contribution and impact of their paper.

There are clearly many things to like about this manuscript. The experimental manipulation is a strong point. So is the theoretical research question as well as the attempt to test theoretical predictions about cognitive dissonance in an applied setting. The study is well-conducted, and the analyses are mostly correct and reported in a straightforward manner. I liked the violin density plots. The findings provide evidence for the idea that emotions and perceptions of election legitimacy polarized over time after the 2020 presidential election.

Thank you for the encouraging comments!

Neither the predictions nor the results are particularly surprising. Most people, even those without any training in social sciences, would predict that losers perceive elections as less legitimate (and experience more negative emotions) than winners, and that this difference increases as the outcome of the elections become more and more clear. Once does not need cognitive dissonance to explain this effect.

Thank you for the comments. We do not have evidence that directly addresses whether the results are particularly surprising to social scientists or to the lay public, as we did not collect such data. We would argue, however, that surprisingness or counter intuitiveness are problematic criteria to evaluate scientific contributions. Excessive weighting of counter intuitiveness might even contribute to perverse incentives. A better criterion, we would argue, is whether the findings advance knowledge of how the world is, surprising or not. Most scientific facts, not surprisingly, are not surprising. 

The latter part of the comment (“Once [sic] does not need cognitive dissonance to explain this effect”) implies there are alternative theoretical explanations of the findings. We are not sure which specific theoretical explanations the reviewer has in mind, but do not dispute their existence. We have therefore taken care to explain that our predictions are derived from, if not exclusively linked to, cognitive dissonance theory (e.g., lines 17-19, 103-105). 

The difference between the two groups, “Declared” and “Undeclared,” is not just that Biden was declared the winner of the elections. Many other things happened between Nov. 4 and 15, 2020. Trump and many Republican elected officials declared the elections as being rigged, whereas Biden and his team kept saying that the elections were legitimate. The observed results could be due to cognitive dissonance reduction, as the authors claim. But they could also be due to respondents simply being influenced by their party leaders or numerous other things that changed between Nov. 4 and 15. Although the study contains an experimental manipulation (= is a “true experiment”) it is unclear what was manipulated here.

Thank you for suggesting this interpretation. This interpretation is broadly consistent with cognitive dissonance theory, and with our suggestion that consumption of polarized media sources contributes to changes in perceived election legitimacy, as discussed in point (2) above. Trump and Biden made different claims about election legitimacy, as did Fox News and NPR. We further suspect that different media sources gave different credence to Trump’s and Biden’s claims of election (il)legitimacy. In both cases, group leaders and political “elites” provided similar dissonance reducing information, with Fox News echoing Trump’s false claims about the election. The moderation results reported in point (1) above implicate the role of consumption of and trust in Fox News versus other sources, but do not strongly differentiate the roles of media versus political leaders. We have acknowledged this overlap in the manuscript (lines 519-521). 

Given the interpretational ambiguities of the results, I feel that the authors overstate their results. I think the authors should adopt more cautious language and delete sentences such as “These findings advance theoretical understanding of polarized perceptions of election legitimacy by more directly implicating rationalizing processes associated with cognitive dissonance and motivated reasoning” and “[…] suggest the role of emotion in the arousal and reduction of dissonance through rationalization.”

We have revised the lines you noted and others to use more cautious language (e.g., lines 480-482). 

Has this study been preregistered? I am asking because the authors made many choices that I would not have made. Here some examples. Party affiliation was measured on a 7-point scale. The authors trichotomized this continuous scale into three categories: Responses 1, 2, and 3 were labeled “Democrat,” response 4 “Independent,” and 4, 6, and 7 “Republican.” Why not a different categorization into responses 1 and 2 (Democrat), 3, 4, and 5 (Independent), and 6 and 7 (Republican)? Or why not treat this variable as a continuous predictor with 7 levels? The latter choice would have made sense because the “quadratic trend” is significant (see Table 1). 

No, the study was not preregistered. 

We appreciate the concern about treating party affiliation as a continuous measure. Following established practice in political science and, to a lesser extent, political psychology, we treat party identification as a categorical measure for two reasons. First, the American National Election Study questions are a series of categorical questions. Respondents are first asked whether they identify as Democrat, Republican, or something else. Democrats and Republicans then indicate whether they identify strongly or moderately. Non-identifiers are asked whether they lean Democrat, lean Republican, or do not identify with either party. These responses are fundamentally categorical. Second, and relatedly, the 1-7 scale above lacks interval properties. The change from 1 (strong Democrat) to 2 (moderate Democrat) is arguably different from the change from 4 (Independent) to 5(lean Republican); the latter represents a qualitative shift in group identity. 

Where does the idea come from to combine all “other” 14 media outlets into one score which is then contrasted to Fox News?

Contrasting Fox News with other sources reflects research from political science that singles out Fox News as a conservative point of comparison (DellaVigna & Kaplan, 2007) and polling research demonstrating that Republicans trust Fox News more than other sources whereas Democrats distrust Fox News (Gramlich, 2020). We have added these references (lines 173, 221). Our selection of the 15 media sources was guided by the availability of their full text, which we use in a different project (lines 277-281).

Given the introduction, I expected the polarizing effect (the increase in difference between Democrats and Republicans between Undeclared and Declared) to be moderated by in-group media consumption. In other words, I expected the authors to test a 3-way interaction between party identification, timing, and media consumption (the latter being a continuous predictor indicating the extent to which respondent consume media known to promote ideas consistent with the respondents’ party identification). I also expected the authors to test a moderated mediation model in which the three-way interactive effect on election legitimacy is moderated by emotions, i.e., the difference between negative and positive emotions. Such a moderated mediation model corresponds to Hayes’ Model #11.

We address this in point (1) above. We report the results of an analysis with higher order interactions that are consistent with our hypotheses (Table 1, line 388-418). We chose not to conduct mediated moderation or moderated mediation analyses given the complexity of higher order 3-way and 4-way interactions and given the exploratory and tentative nature of such mediation analyses. 

Minor points:

I was confused by the fact that different terms were used interchangeably. “Party identification” was sometimes called “party identity” and “partisan ID.” “Election legitimacy” was sometimes referred to as “vote legitimacy,” “perception of legitimacy,” and “confidence in vote legitimacy.” “Media outlets” are also “news outlets” and “sources.” “Media trust and consumption” is also called “engagement.” “National votes” are sometimes “nationwide votes.”

Thank you for pointing out these inconsistencies. We have resolved them. 

I didn’t understand the sentence “Respondents answered both questions on non-numeric scales” (p. 13). Are the 5-point scales, which were mentioned two sentences before, non-numeric? A similar issue occurs on page 13 where the authors say “Participants answered on two scales presented without numbers (1 = Not at all confident; 5 = Very confident).” Do the authors mean to say that they presented respondents with five verbal labels and that they later assigned the numbers 1 to 5 to these labels?

We have clarified that respondents did not see numerical labels. We assigned numerical labels for data analyses (lines 258-259, 266-267, 284-285).

The choice of the 15 media outlets is surprising. How come the authors did not include a larger number of conservative outlets?

As noted above, our selection of media outlets was guided by the availability of full text databases for a different project. We acknowledge there are additional conservative media outlets that were not included in the study (lines 277-281, 509-510), while also recognizing that Fox News, as noted above, has been singularly identified in the political science literature as a strong conservative voice among mainstream media (lines 172-175). 

It is not very informative to compute for each respondent a correlation between trust and consumption across the 15 media outlets, i.e., 1236 correlations each with an N of 15 (p. 13-14). It would be better to compute for each media outlet a correlation across all participants, i.e., 15 correlations each with an N of 1236. The authors should then report the median and the range of these 15 correlations.

The two correlations address slightly different associations, with convergent evidence bolstering our decision to average ratings of trust and consumption. At the individual level, we compute the correlation between 15 pairs of trust and consumption ratings, which we then average across 1236 respondents (average within-person r = .51). This correlation indicates that, on average, individuals report that they consume more of those media sources that they also rate as more trustworthy, and that this association occurs, on average, across people (lines 285-287). At the level of media outlet, we compute the correlation between pairs of trust and consumption ratings across 1236 respondents, given data availability. We then average those 15 correlations (average between-person r = .51, range: [.36, .67]). This correlation indicates that the more respondents trust a particular media source, the more they report consuming that source, and that this association occurs, on average, across the 15 sources (lines 287-293). 

On page 14, the authors report a one-df test, F(1, 1078) = 3.13, p = .077, which is a 2-df test.

This was correctly, but unclearly, reported as a 1-df test of the interaction of timing (Undeclared, Declared) and partisanship (Democrat, Republican), with Independents excluded. We now report the 2-df test of the interaction between partisan identification and timing for clarity (lines 302-303).

I’d drop the factor “vote type” in the analyses reported on pages 14-15 and I’d collapse the top and bottom panels in Figure 2). The factor doesn’t add anything to the paper. Given that the two vote confidence ratings are averaged in the remaining analyses, including it as a factor in the earlier analyses creates confusion for readers. The fact that the party identification by timing interaction was stronger for national votes can be mentioned in a footnote.

Thank you for raising this point. Following your suggestion, we combined the two measures of perceived election legitimacy in the primary analysis (lines 307-323) and in Figure 2, consistent with previous research (Sances & Stewart III, 2015). Our theorizing does not directly imply a difference between these factors. 

We disagree, however, that examining the factors separately does not add anything to the paper. Previous research has found systematic differences such that people perceive the health and functionality of democracy at national level to be lower than at the personal level (Sances & Stewart III, 2015). Our study speaks to these differences, so we report analyses at the end of the Results section that includes vote type as a factor (lines 428-447). We speculate in the Results and Discussion that confidence in national vote counting may be more labile in response to motivated reasoning than confidence in own vote counting (lines 443-447, 517-522). 

I did not understand right away what the authors meant by “systematic legitimacy of national votes” (p. 15).

For clarity, we have rewritten the introduction to the Perceived Election Legitimacy of Own Vote and Nationwide Vote section to clarify the point we are trying to make with our previous poorly worded statement (lines 428-431).

It says on page 15 that “Republicans’ confidence that their own votes were correctly counted did not significantly decrease over time (MUndeclared = 3.31, […] MDeclared = 3.31)”, but then the red line in the top middle panel in Figure 2 is not perfectly flat. How is this possible?

Thank you for catching this mistake! The correct means are MUndeclared = 3.35 and MDeclared = 3.24, which we corrected in the manuscript. 

Figure 2: I’d put the violin density plots for the Independents in the middle rather than on the right side.

Thank you for the suggestion. We have decided to keep the figure in its current state with the order remaining Democrat, Republican, and Independent because our primary comparison of interest is between Democrats and Republicans. In its current order it is easier to visually compare group changes in emotions over election phases when they are placed side-by-side.

The authors say on page 17 “In a regression analysis, negative emotions predicted lower confidence that votes were counted correctly,” but it is not clear what confidence ratings they are referring to, the “own vote,” the “nationwide vote,” or the average of the two ratings. The same issue exists in the title of Table 1 and the analyses reported on p. 20.

We have changed the terminology from “confidence that votes were counted correctly” to “perceived election legitimacy” which is defined as the mean of “nationwide vote confidence” and “own vote confidence” to clarify our analyses (lines 261-266). 

It is incorrect to say that Figure 4 is a graphic representation of the result that “Democrats’ and Republicans’ differential engagement with polarized media corresponded with their polarized confidence in vote legitimacy” (p. 19). Confidence in vote legitimacy (= perceived election legitimacy) is not shown in Figure 4.

Thank you for pointing this out. We have clarified that Figure 4 presents media perceptions, without the presentation of perceived election legitimacy (lines 369-370).

The significant “curvilinear trend” (b = 0.16) reported in Table 1 is surprising. This coefficient and Figure 2 suggest that the polarizing effect is mostly due to Democrats becoming more confident. Is this finding consistent with the authors’ theoretical analyses? Wouldn’t we expect cognitive dissonance effects be strongest for losers?

We partially addressed this comment above. It is true that the magnitude of Democrats’ increase in perceived election legitimacy is greater than the magnitude of Republicans’ decrease in perceived election legitimacy. However, interpreting that difference is complicated because the conclusions are not symmetric. For Democrats, consistency motivations and the emerging evidence of correctly counted votes point in the same direction; for Republicans, consistency motivations and the emerging evidence of correctly counted votes point in opposing directions. The combination of these effects imply a larger effect for Democrats than for Republicans. We acknowledge this issue in the Discussion (lines 463-476). 

I don’t understand why the authors first include media outlet (= source) as a within-subject factor (which is identical to computing a difference score; see p. 18) but then run analyses in which they include both media outlet scores as predictors (Table 1). Is type of media outlet hypothesized to be a moderator, or are the authors predicting the existence of two (parallel?) mediators?

We analyze ratings of media outlets two ways. First, to capture mean differences between groups, we report a 3(partisan identification: Democrat, Republican, Independent) × 2(media outlet: Fox News, Other Outlets) ANOVA. As you note, the media outlet factor is equivalent to a difference score, and the partisan identification by media outlet interaction reflects the fact that, on average, Republicans and Democrats trust and consume different media sources (Figure 4).

Second, in the moderation analyses (Table 1), we enter ratings of Fox News and other outlets as separate predictors, each interacting with the categorical contrast coded predictors and with each other. This is important because the ratings of Fox News and other outlets are slightly positively correlated at the individual level but negatively related at the mean level. To illustrate, in a series of regressions predicting ratings of Fox News, the effect of ratings of other outlets is not-significant when it is the only predictor (b = 0.01, t(1211) = 0.19, p = .847) but becomes significantly positive (b = 0.33, t(1208) = 7.63, p < .001) when the regression also includes the two contrast codes representing participant partisan identification (Democrat vs. Republican b = 1.28, t(1208) = 15.90, p < .001; Independent vs. Democrat and Republican b = 0.28, t(1208) = 3.09, p = .002). An analogous pattern occurs when predicting ratings of the 14 other sources. In other words, the ratings of Fox News and other sources present something of a Simpson’s paradox (https://en.wikipedia.org/wiki/Simpson%27s_paradox). Given this, it would be inappropriate to analyze ratings of Fox New and Other Outlets as a fixed-effect within person factor in the moderation models. 

The sentence “It is noteworthy how in essays …” (p. 22) is formulated in an awkward manner.

We have rephrased the sentence for clarity (line 489-491).

The authors dedicate four paragraphs to the types of questions that future research might examine (pages 23-25). I think these ideas can be reduced to one paragraph. It would be more interesting for the authors to discuss the implications of their findings rather than provide a list of the numerous things they didn’t do in the present research.

We have slightly condensed our suggestions for future research to two paragraphs (lines 517-535). We are hesitant to condense the discussion further because our directions for future research accompany our acknowledgements of shortcomings in the present experiment. We appreciate the suggestion that we discuss the implications of our findings and have expanded the paragraph where we do so (lines 530-535).

References

DellaVigna, S., & Kaplan, E. (2007). The Fox News effect: Media bias and voting. The Quarterly Journal of Economics, 122(3), 1187-1234.

Gramlich, J. (2020, August 18). 5 facts about Fox News. Pew Research Center. https://www.pewresearch.org/fact-tank/2020/04/08/five-facts-about-fox-news/. 

Sances, M. W., & Stewart III, C. (2015). Partisanship and confidence in the vote count: Evidence from US national elections since 2000. Electoral Studies, 40, 176-188.

---

## [Editor Report · Decision Letter 1]

20 Oct 2021

When Election Expectations Fail: 

Polarized Perceptions of Election Legitimacy Increase with Accumulating Evidence of Election Outcomes and with Polarized Media

PONE-D-21-23399R1

Dear Dr. Van Boven,

We’re pleased to inform you that your manuscript has been judged scientifically suitable for publication and will be formally accepted for publication once it meets all outstanding technical requirements.

Thank you for the diligent work in responding to the review team's comments/concerns. This was an excellent revision!

Kind regards,

Jeff Galak, PhD

Academic Editor

PLOS ONE
---

## [Editor Report · Acceptance letter]

8 Nov 2021

PONE-D-21-23399R1 

When Election Expectations Fail:
Polarized Perceptions of Election Legitimacy Increase with Accumulating Evidence of Election Outcomes and with Polarized Media 

Dear Dr. Van Boven:

I'm pleased to inform you that your manuscript has been deemed suitable for publication in PLOS ONE. Congratulations! Your manuscript is now with our production department. 

Kind regards, 

on behalf of

Dr. Jeff Galak 

Academic Editor

PLOS ONE